

# Identification of pivotal lncRNAs in papillary thyroid cancer using lncRNA–mRNA–miRNA ceRNA network analysis

Weiwei Liang[1,*] and Fangfang Sun[2,3,*]

[1] Department of Endocrinology, The Second Affiliated Hospital, Zhejiang University School of Medicine, Hangzhou, China
[2] Department of Colorectal Surgery, The Second Affiliated Hospital of Zhejiang University School of Medicine, Hangzhou, China
[3] Cancer Institute (Key Laboratory of Cancer Prevention and Intervention, Chinese National Ministry of Education; Key Laboratory of Molecular Biology in Medical Sciences, Zhejiang Province, China), The Second Affiliated Hospital of Zhejiang University School of Medicine, Hangzhou, China
* These authors contributed equally to this work.

## ABSTRACT

**Background:** To identify pivotal lncRNAs in papillary thyroid cancer (PTC) using lncRNA–mRNA–miRNA ceRNA network analysis.

**Methods:** We obtained gene expression profiles from the gene expression omnibus database. Cancer specific lncRNA, cancer specific miRNA and cancer specific mRNA were identified. An integrated analysis was conducted to detect potential lncRNA–miRNA–mRNA ceRNA in regulating disease transformation. The lncRNA regulated gene ontology (GO) terms and regulated pathways were performed by function analysis. Survival analysis was performed for the pivotal lncRNAs.

**Results:** A total of four lncRNAs, 15 miRNAs and 375 mRNAs are identified as the key mediators in the pathophysiological processes of PTC. GO annotation enrichment analysis showed the most relevant GO terms are signal transduction, integral component of membrane and calcium ion binding. Kyoto Encyclopedia of Genes and Genomes pathway enrichment analysis showed different changed genes mainly enriched in pathways in cancer, PI3K-Akt signaling pathway and focal adhesion. Among four lncRNAs, only SLC26A4-AS1 was significantly associated with PTC patient disease free survival.

**Conclusion:** This study has constructed lncRNA–mRNA–miRNA ceRNA networks in PTC. The study provides a set of pivotal lncRNAs for future investigation into the molecular mechanisms.

Corresponding author
Weiwei Liang,
helenliangww@zju.edu.cn

## INTRODUCTION

Most of the human genome transcribe RNAs that do not code for proteins. These non-coding RNAs (ncRNAs) affect normal expression of the genes, including oncogenes and tumor suppressive genes, which make them a new class of targets for study in tumorigenesis. NcRNAs include microRNA (miRNA), long non-coding RNA (lncRNA), circular RNA et al. MiRNA are highly conserved non-coding RNAs of approximately

21–24 nucleotide, which can interact with target mRNAs to regulate the expression of the gene. Studies have shown that miRNAs are involved in the initiation and progression of cancers (*Hayes, Peruzzi & Lawler, 2014*), and miRNA-targeted therapeutics have already reached clinical development (*Slaby, Laga & Sedlacek, 2017*). Recently, the importance of lncRNA is gradually realized. LncRNA is a group of non-coding RNAs (lncRNAs) that are more than 200 base pairs. LncRNAs have diverse role in genetic regulation. Recent studies have demonstrated that lncRNA play a major regulatory role in tumorigenesis (*De Oliveira et al., 2018*).

The competing endogenous RNA (ceRNA) hypothesis, presented by *Salmena et al. (2011)*, has been described as the "Rosetta Stone" for decoding the RNA language used in regulating RNA crosstalk and modulating biological functions. RNAs regulate each other with miRNAs response elements and this mechanism is known as "ceRNA" hypothesis. Many studies have suggested miRNA-mediated ceRNA regulatory mechanisms play crucial roles in the initiation and development of tumors (*De Oliveira et al., 2018*).

Papillary thyroid cancer (PTC) is the most common type of thyroid cancer. PTC accounts for about 80% of all thyroid cancers (*Carling & Udelsman, 2014*). It is the fifth most common cancer in women. The incidence of thyroid cancer in the United States and worldwide has increased 300% over the past three decades, due predominantly to an increase in PTC (*Siegel, Miller & Jemal, 2017*). One study predicts that PTC will become the third most common cancer in women at a cost of $19–21 billion in the USA in 2019 (*Aschebrook-Kilfoy et al., 2013*). Although PTC generally has a good prognosis, 10% patients suffer from local recurrence and/or distant metastasis. Thus, it is essential to clarify the mechanisms of PTC and identify latent biomarkers. Recently, there is an increasing number of lncRNAs discovered in tumorigenesis of PTC (*Nikiforov, 2017*; *Lei, Gao & Xu, 2017*). *Feng et al. (2019)* found lncRNA n384546 promotes thyroid papillary cancer progression and metastasis. *Liang et al.'s (2019)* recent study showed LncRNA MCM3AP-AS1 promotes proliferation and invasion through regulating miR-211-5p/SPARC axis in PTC. But most of the researches focus on one particular lncRNA, our knowledge about the lncRNA–mRNA–miRNA ceRNA network of PTC is still limited.

In this study, we have taken full advantage of the rich data from the gene expression omnibus (GEO) database. The aberrant expression profiles of lncRNA, mRNA, and miRNA in PTC were filtered. The lncRN-miRNA-mRNA ceRNA network was constructed. Four lncRNAs were identified as pivotal lncRNAs. This paper represented a significant leap forward in understanding the biological functions of lncRNAs in PTC.

## MATERIALS AND METHODS

### Data sources

We conducted a search of the GEO database (www.ncbi.nlm.nih.gov/geo/) for high-throughput genomics experiments of papillary thyroid carcinoma. We used the following search terms: thyroid tumor, thyroid carcinoma, papillary thyroid carcinoma, papillary carcinoma of thyroid. The RNA sequencing expression data of three datasets (GSE33630, GSE3467, GSE3678) were obtained from GEO. DESeq2 package (*Love, Huber & Anders, 2014*) was used to identify different expressed genes. The quality of gene expression data was analyzed and visualized using the ggplot2 package of R software for each group and sample.

## Identification of cancer specific lncRNA, cancer specific miRNAs, cancer specific mRNA

The analysis of differentially expressed genes was carried out by limma package (*Ritchie et al., 2015*), which includes lmFit, eBayes, and topTable functions. $p < 0.05$ and abs(log2) fold change > 1 were used as the cut-off criteria. Commonly different changed genes (DEGs) from the datasets were integrated using Venn analysis. LncRNA, mRNA, and miRNA were identified by bioMart package (*Durinck et al., 2005*).

## Construction of lncRNA–miRNA–mRNA network of PTC

The lncRNA–miRNA–mRNA network of each cancer was constructed by following steps:

1. Prediction of miRNA targeted lncRNA: we used DIANA-LncBase v2 (http://carolina.imis.athena-innovation.gr/) (*Paraskevopoulou et al., 2016*), starBase (http://starbase.sysu.edu.cn/) (*Li et al., 2014*; *Yang et al., 2011*) and RAID v2.0 (http://www.rna-society.org/raid/) (*Yi et al., 2017*) to analyze interactions between lncRNAs and miRNAs;
2. Identification of miRNA targeted mRNA: we used DIANA-TarBase v8 (http://carolina.imis.athena-innovation.gr/) (*Karagkouni et al., 2018*) and TargetScan (http://www.targetscan.org/) (*Agarwal et al., 2015*) to predict miRNA-targeted mRNA; and
3. LncRNA–miRNA–mRNA network construction and visualization: Cytoscape 3.7.0 software (*Shannon et al., 2003*) was utilized to construct and visualize ceRNA network as well as its subnetwork based on individual lncRNA.

## Functional analysis

Gene ontology analysis (GO) was used to identify characteristic biological attributes. Kyoto Encyclopedia of Genes and Genomes pathway (KEGG) (*Kanehisa et al., 2017*) enrichment analysis was performed to identify functional attributes. We use search tool for the retrieval of interacting gene (*Szklarczyk et al., 2017*), ClueGo (*Bindea et al., 2009*) and Cluepedia (*Bindea, Galon & Mlecnik, 2013*) to access GO and pathway enrichment analysis. $p < 0.05$ was set as the cut-off criterion.

## Survival analysis of pivotal lncRNAs

The lncRNAs in PTC lncRNA–miRNA–mRNA network were indentified as pivotal lncRNAs. LncRNAs correlations with patient survival were featured in GEPIA (*Tang et al., 2017*). Available the cancer genome atlas (TCGA) patient survival data were used for Kaplan–Meier survival analysis and to generate overall and disease-free survival plots.

## RESULTS

Three GEO datasets (GSE3678, GSE3467, GSE33630) were obtained in our study: Three datasets were all based on GPL570 platform (Affymetrix Human Genome U133 Plus 2.0 Array). The GSE3678 dataset had seven PTC samples and seven paired normal thyroid

tissue samples. GSE3467 included nine PTC samples and nine paired normal thyroid tissue samples. GSE33630 had 49 PTC samples and 45 normal thyroid tissue samples.

The normalized expression values (ranging from 0 to 20 log2 (TPM+1)) and the distributions were similar between the three groups in each dataset (Fig. 1). The principal component showed that samples were easily grouped into different groups in each dataset (Fig. 1). Based on these data distribution analyses, further bioinformatics analyses could be performed based on the available data. Following data processing using LIMMA, 1189, 1194 and 2226 differentially expressed genes from the expression profile datasets GSE3678, GSE3467 and GSE33630 were extracted, respectively (Figs. 2A–2D). A total of 615 consistently expressed genes (Table S1), including four lncRNAs, were identified by Venn analysis (Fig. 2D). Hierarchical clustering of the identified DEGs is displayed as a heatmap in Fig. 2F.

The relationships between cancer specific lncRNAs and cancer specific miRNAs were first evaluated. We obtained 96 abnormally expressed miRNAs significantly relevant to PTC survival from OncomiR. Four PTC-specific lncRNAs putatively interacted with 15 PTC-specific miRNAs. Then, the relationships between cancer specific miRNAs and cancer specific mRNAs were evaluated. A total of 375 PTC-specific mRNA targeted by 15 PTC-specific miRNAs were identified. Based on the above data, the lncRNA–miRNA–mRNA ceRNA network of PTC (Fig. 3) was established and plotted using Cytoscape 3.7.0.

We analyzed the mRNAs of the lncRNA–miRNA–mRNA ceRNA network and identified the lncRNA regulated GO terms (Fig. 4; Tables S2–S4). In PTC lncRNA–miRNA–mRNA ceRNA network, the result showed that the mRNAs related to biological process were most relevant to signal transduction. mRNAs related to cellular component were most relevant to integral component of membrane. mRNAs related to molecular function were most relevant to calcium ion binding.

We performed pathway enrichment analysis of the mRNAs of the lncRNA–miRNA–mRNA ceRNA network and identified the lncRNA regulated pathway (Fig. 5; Table S5). Functional analysis showed the top 3 lncRNA regulated pathway are pathways in cancer, PI3K-Akt signaling pathway, focal adhesion.

We used GEPIA to perform survival analysis of pivotal lncRNAs (Fig. 6). We generated SLC26A4-AS1, NR2F1-AS1 and TNRC6C-AS1's data with patients' survival information. Only SLC26A4-AS1 was significantly associated with PTC patient disease free survival ($p = 0.002$).

## DISCUSSION

The last few years have nominated ceRNA hypothesis as a novel layer of gene regulation. Our knowledge about the molecular mechanism of lncRNA in cancer is still limited. In the present study, we identified cancer specific lncRNAs, miRNAs, and mRNAs in PTC. According to the bioinformatics differential analysis we constructed a ceRNA network. We predicted functions of differentially expressed genes in PTC by GO and pathway analysis. Pivotal lncRNAs (SLC26A4-AS1, NR2F1-AS1, TNRC6C-AS1, LOC646736) from the ceRNA network were further investigated for their correlations with survival based on RNA sequencing profile from TCGA.

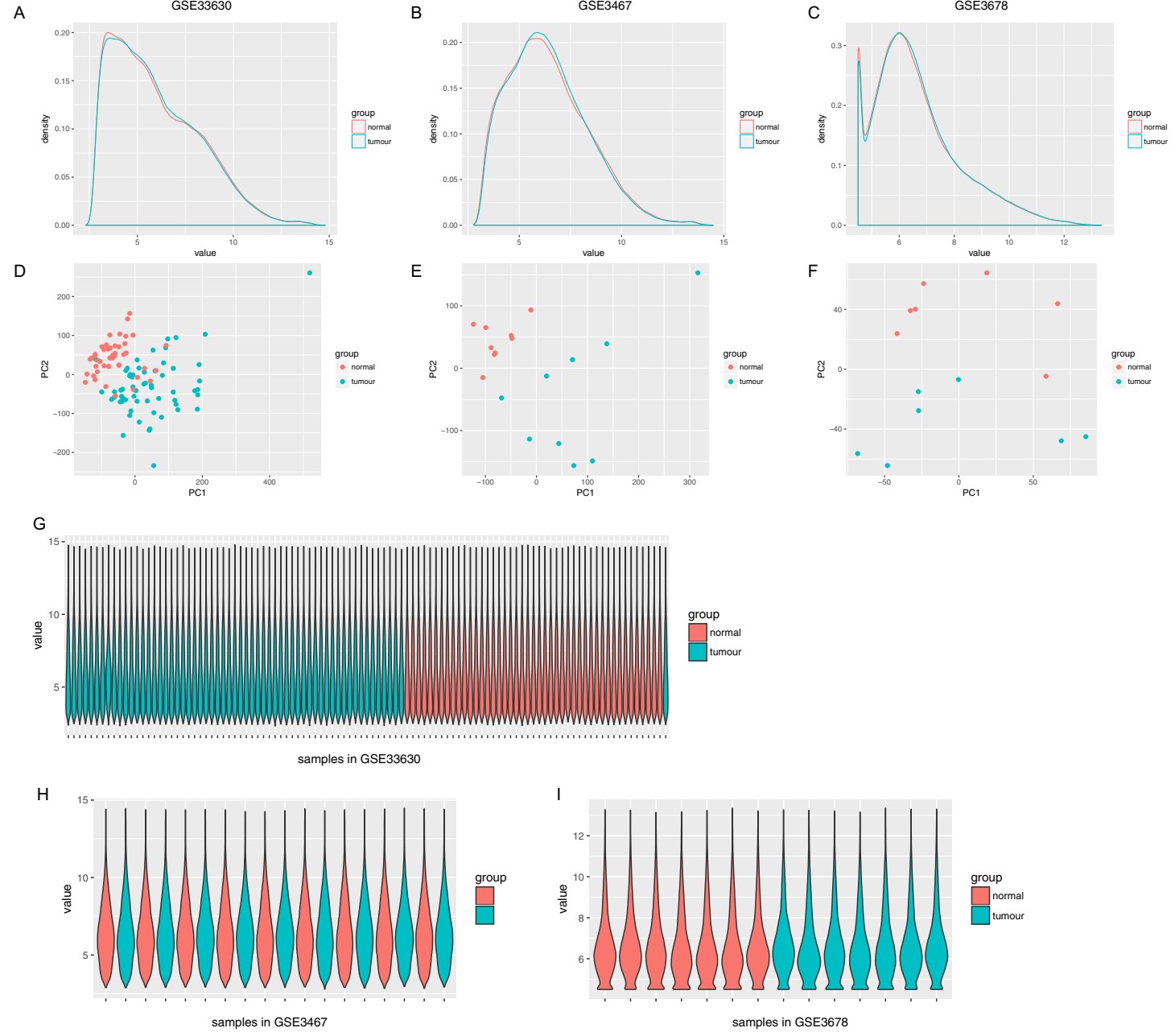

**Figure 1 Distribution analysis of gene expression.** (A–C) Distribution of gene expression levels in each group, where the *x*-axis indicates the log2 value (TPM+1) and the *y*-axis shows the proportion of genes. (D–F) Principle component analysis of each sample. Dots represent the PC value of each sample. (G–I) Distribution of genes based on the expression level of each sample, where *x*-axis shows each analyzed sample and *y*-axis shows the gene value distribution in each sample.

Of the genes that were involved in the ceRNA network, KEGG analysis showed that these genes were mainly enriched in two pathways related with cancer: pathways in cancer and the PI3K-Akt signaling pathway. There was evidence showed that the PI3K-Akt signaling pathway plays an important role in PTC tumorigenesis (*Cancer Genome Atlas Research Network, 2014*). *Hao et al.'s (2019)* study showed that PI3K-Akt signaling

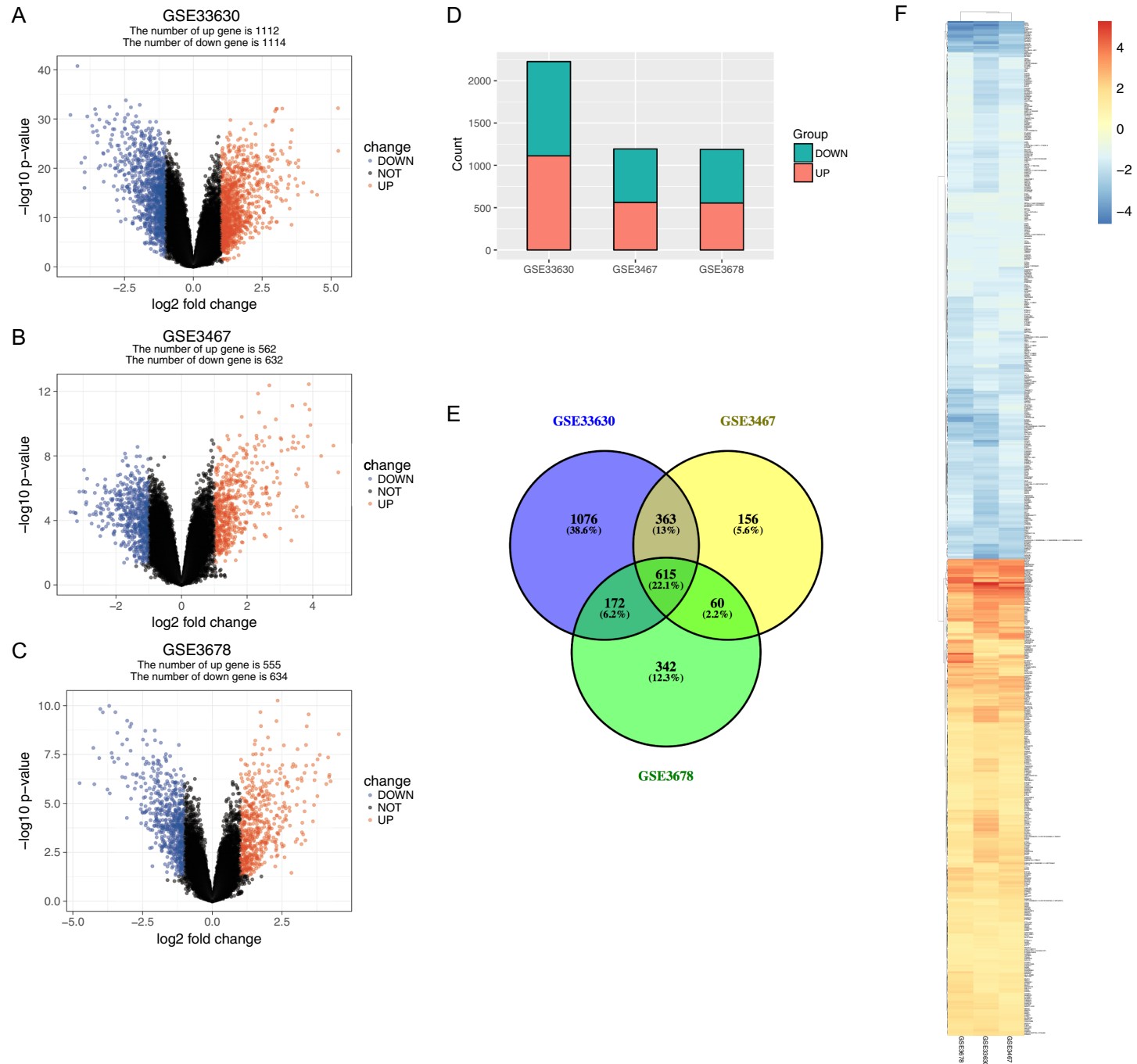

**Figure 2 Gene expression of each dataset.** (A–C) Volcano graph displaying pairs of expressed genes. Red dots indicate significantly up-regulated differential expressed genes and blue dots indicate significantly down-regulated DEGs that passed the screening threshold. (D) Differential expressed gene counts of each dataset. (E) Venn analysis among DEGs of three datastes. (F) Heatmap and hierarchical clustering of identified DEGs. Up- or down-regulated genes are colored in red or blue.

pathway activation was associated with PTC cell proliferation, migration, and invasion. *Zheng et al. (2018)* found TEKT4 promoted PTC cell metastasis through activating PI3K/Akt pathway. The regulation function of lncRNAs through PI3K/Akt pathway in PTC is

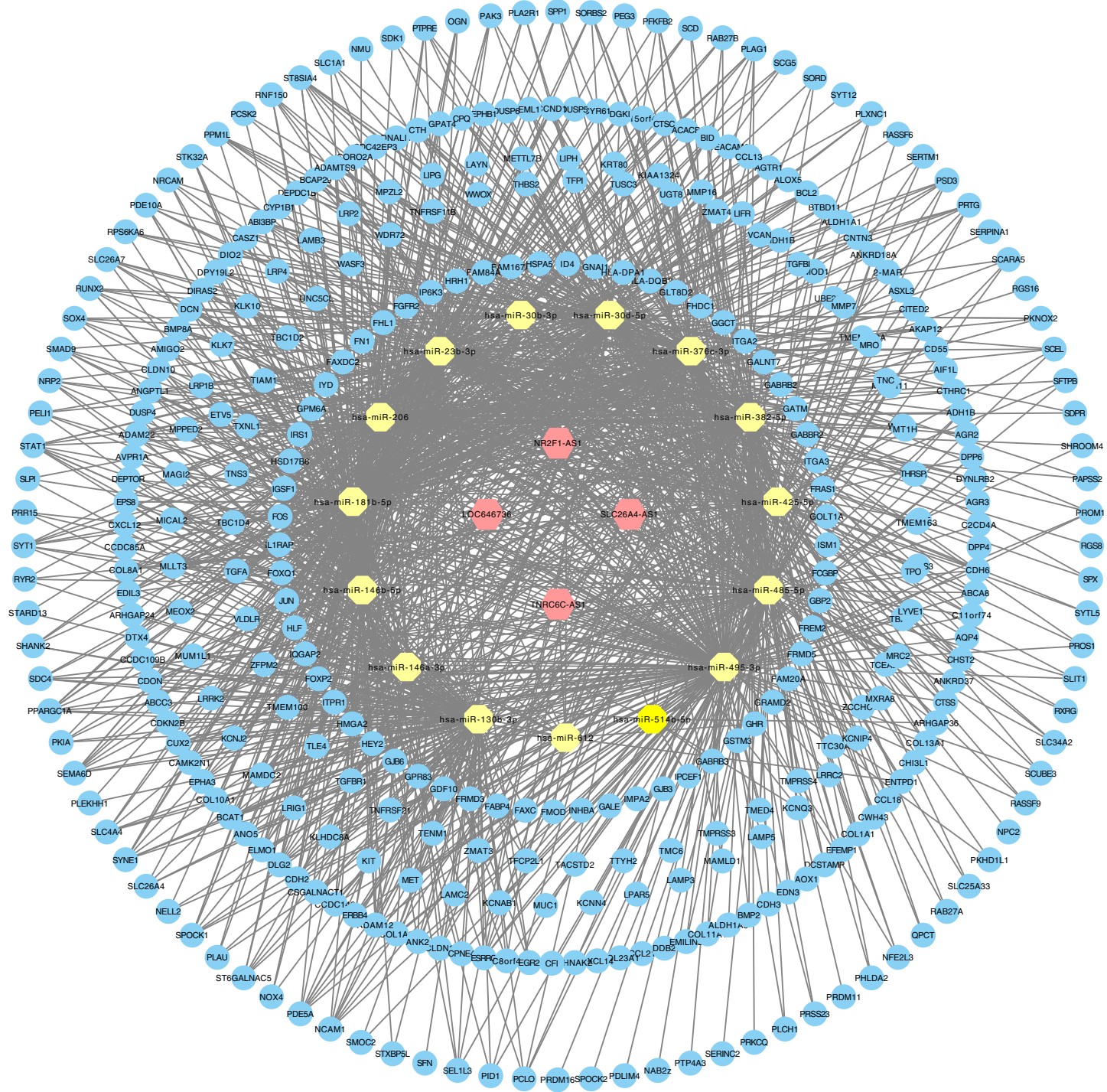

**Figure 3 LncRNA–miRNA–mRNA ceRNA network in PTC.** Hexagon represented cancer specific lncRNA. Octagon represented cancer-specific miRNA. Ellipse represented cancer-specific mRNA.

worth study. There were only two studies published focused on the regulation function of lncRNAs through the PI3K/Akt pathway in PTC. *Wen et al. (2019)* found that lncRNA ABHD11-AS1 promotes tumor progression in PTC by regulating the PI3K/Akt signaling

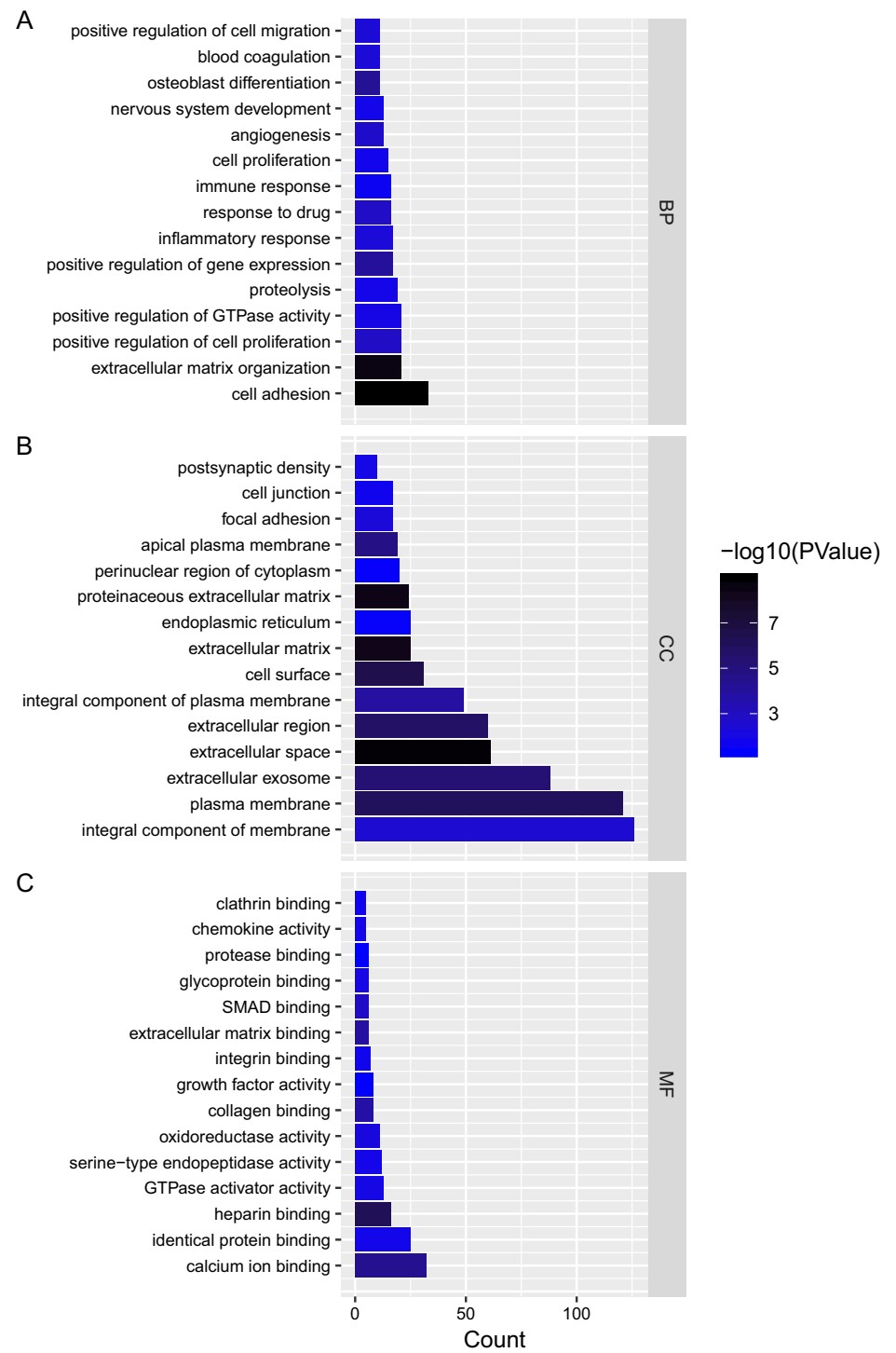

**Figure 4 Gene ontology analysis of mRNAs in ceRNA network.** (A–C) Gene ontology analysis of mRNAs in ceRNA network. *x*-axis reflects gene count; *y*-axis reflects different GO terms. The column color reflects *p*-value ($-\log10(p\text{-value})$): black represents the biggest value, blue represents the smallest value.

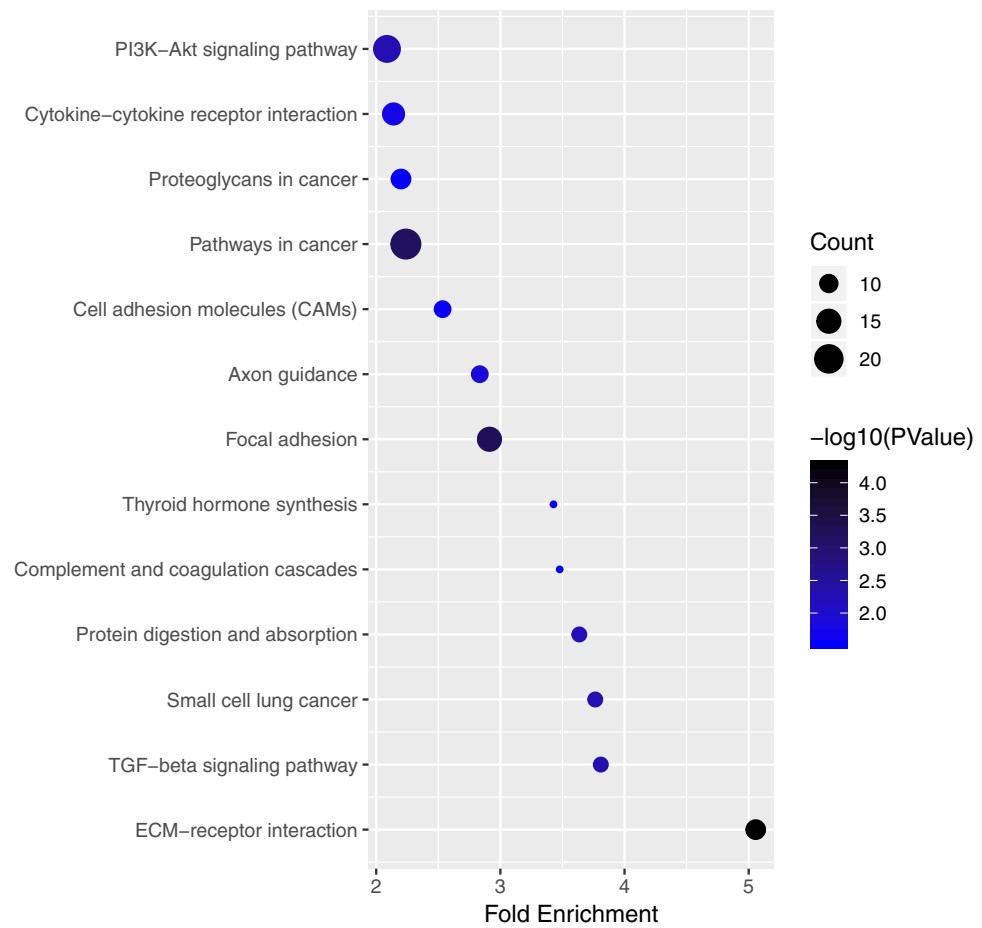

**Figure 5 Significantly enriched pathway terms of mRNAs in ceRNA network.** The node size reflects gene count: the bigger the gene count, the bigger the node size is. The node color reflects *p*-value (−log10(*p*-value)): the bigger the −log10(*p*-value) value, the darker the node color is.

pathway. *Liu et al. (2019)* showed downregulation of NEAT1 reversed the radioactive iodine resistance of PTC cell via the miR-101-3p/FN1/PI3K-AKT signaling pathway. The ceRNA network identified in our study provided useful clues of further study.

Among four pivotal lncRNAs, SLC26A4-AS1 is associated with PTC disease free survival. *Fagerberg et al.'s (2014)* study showed SLC26A4-AS1 is restricted expressed in thyroid. Our understanding of SLC26A4-AS1's function is limited. Further study is worthwhile to detect SLC26A4-AS1's role in PTC.

Long non-coding RNA NR2F1-AS1 is broadly expressed in brain, gall bladder and other tissues. There is some evidence indicating that lncRNA NR2F1-AS1 are associated with other cancer types. *Huang et al.'s (2018)* study showed lncRNA NR2F1-AS1 regulates hepatocellular carcinoma oxaliplatin resistance via miR-363. *Wang, Zhao & Mingxin (2019)* recent study showed that lncRNA NR2F1-AS1 is involved in the progression of endometrial cancer. Studies of lncRNA NR2F1-AS1 in PTC is limited, so they are worthy of future research.

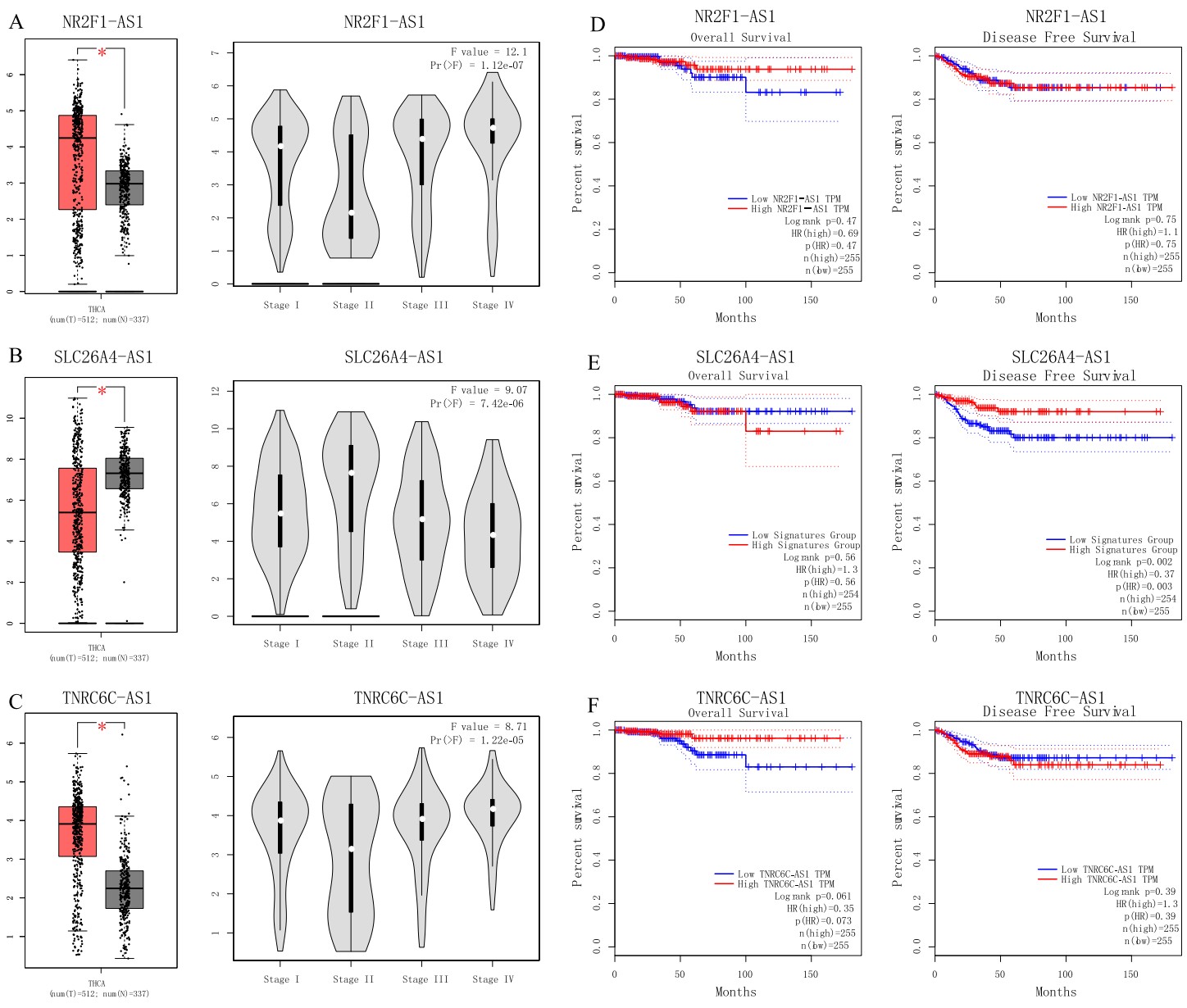

**Figure 6 (A–C) Analyses of pivotal lncRNAs expression in tumor vs. normal tissues and in different tumor stages. (D–F) Overall and disease-free survival analyses of hub lncRNAs.** (A–C) Analyses of pivotal lncRNAs expression in tumor vs. normal tissues and in different tumor stages. The red box represents tumor tissue; gray box represents normal tissue; and dots presents each sample value. $^*p < 0.05$; other $p$-values are shown on the diagrams. (D–F) Overall and disease-free survival analyses of hub lncRNAs. $p$-values are shown on the diagrams.

TNRC6C-AS1 is broadly expressed in lymph node, spleen and other tissues. *Hou et al. (2018)* showed that lncRNA TNRC6C-AS1 regulates UNC5B in thyroid cancer to influence cell proliferation, migration, and invasion as a ceRNA of miR-129-5p. In H9c2 cells, miR-129-5p inhibits autophagy and apoptosis induced by hydrogen peroxide via the PI3K/Akt/mTOR signaling pathway (*Zhang, Zhang & Zhang, 2018*). So, we hypothesis

TNRC6C-AS1 is an important lncRNA in the regulation of PI3K-Akt signaling pathway in tumorigenesis of PTC.

LOC646736 is restricted expressed in thyroid. Our study is consisted with Xu et al.'s (2014) result. The function of LOC646736 in tumorigenesis is unknown. Further molecular biological experiments are required to confirm the function of the identified genes.

In our study, we identified four pivotal lncRNAs in PTC. Among them, SLC26A4-AS1 is associated with PTC disease-free survival. It is meaningful to explore new biomarkers for diagnosis, prognostication, and therapeutic targets of PTC so as to develop more effective surveillance and treatment programs. Although we had limited knowledge about SLC26A4-AS1, this study provides a good direction for further study. Functional studies that further delineate the biologic basis of the PTC are needed. Further studies with much larger sample sizes will be needed.

## CONCLUSION

In conclusion, our data provide a comprehensive bioinformatics analysis of pivotal lncRNAs in PTC. These lncRNAs are worthwhile for further study as novel candidate biomarkers for the clinical diagnosis of PTC and potential indicators for prognosis.

### Funding
The authors received no funding for this work.

### Competing Interests
The authors declare that they have no competing interests.

### Author Contributions
- Weiwei Liang conceived and designed the experiments, performed the experiments, contributed reagents/materials/analysis tools, prepared figures and/or tables, authored or reviewed drafts of the paper, approved the final draft.
- Fangfang Sun analyzed the data, contributed reagents/materials/analysis tools, prepared figures and/or tables.

### Data Availability
Data is available at NCBI GEO: GSE33630, GSE3467, GSE3678.

### Supplemental Information
Supplemental information for this article can be found online at http://dx.doi.org/10.7717/peerj.7441#supplemental-information.

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
