# Peer review of "Identification of pivotal lncRNAs in papillary thyroid cancer using lncRNA–mRNA–miRNA ceRNA network analysis"

_PeerJ, doi:10.7717/peerj.7441_

## Round 0.1 · original submission · Minor Revisions

The reviewers disagree (as you will see) on the overall decision, but I am of the opinion that minor revisions will be sufficient for this manuscript to reach the required standard. Please make every effort to address all the pints raised by the reviewers. Reviewer 1 in particular feels you need to elaborate your discussion of the data further, and I am minded to agree. Most of the comments of Reviewer 2 are straightforward to address.

I would add that the figures need a little work. Figure 3 needs to be re-drawn, and in many figures the text use is either hidden behind the graphs (Figure 6) or too small to read clearly (e.g;. right side of Figure 5).

I also ask that you include more depth on the statistical analysis performed. Some further clarity here would be useful for others.

Reviewer 1 ·

Basic reporting

I suggest that you improve the first sentence of your Introduction (line 45). Using “most of the human genome transcribes RNAs that do not code for proteins” sounds more appropriate.
On line 47, the authors say that “NcRNAs can be divided into two classes: small ncRNAs and long ncRNAs (lncRNAs)”. This affirmation is over simplified and I suggest a better definition, looking in the innumerous reviews about ncRNAs
On line 58 the citation is not in agreement with the idea. Double check the references
The English language should be improved to ensure that a better understand your text. Review your writing, including lines 62, 65, 81, 141– the current phrasing makes comprehension difficult.
Line 63 – the bibliographic review is out of date. A quick search in PubMed shows 19 papers in 2019 with regard to PTC and lncRNA.

Experimental design

“The RNA sequencing expression data of 3 PeerJ reviewing PDF | (2019:04:36581:0:1:NEW 15 Apr 2019) Manuscript to be reviewed 74 datasets (GSE33630, GSE3467, GSE3678) were obtained from GEO”
All the datasets were generated through array.

“DESeq2 package[10] was used to normalize 75 read counts of gene expression in two GEO datasets”
DESeq2 was used to normalize array data?

“Construction of lncRNA-miRNA-mRNA network of PTC”
For the construction of the ceRNA network no statistical method was used to evaluate the expression ratio of lncRNA-miRNA-mRNA. Only direct prediction of target is not enough to establish a ceRNA network.

Validity of the findings

Figure 3: This network representation is very poorly represented. Cannot identify nodes or edges of the network.

Line 151 – The authors write “Therefore, our results suggested that NR2F1-AS1 may play an important role in the progression and development of PTC and the cancer genes related pathways”. There is no data in this study to support this assumption.

Additional comments

The idea of the paper is good, but not well elaborated. There are language mistakes, references could be more updated and the analysis is poor. Nice figures, but without proper interpretation, mean nothing.

·

Basic reporting

- The authors have used a clear, ambiguous language, some editing is required.
- Sufficient background of microRNAs and lnRNAs is provided.
- Additional background about PTC can be included. Like statistics (deaths... or incidence) to justify why this disease was chosen for this study.
- The authors have shared necessary figures, tables and raw data.
- The results are relevant to the hypothesis and are clearly stated.

Experimental design

- The research findings are original and in line with the scope of the journal.
- Hypothesis is unique and stated. The authors investigate lcRNAs in papillary thyroid cancer using comprehensive network analysis techniques.
- They also use good techniques (Both Gene Ontology annotation and KEGG pathway enrichment analysis) to prove their point.
- Methods are described well and are comprehensive as well.
- The investigation seems to of high technical standard.

Validity of the findings

- The authors state that 4 lncRNAs were identified as pivotal lncRNAs in in PTC. These are novel findings and also very relevant to the field.
- Additionally, co-relation of lncRNAs with Survival has been shown.
- They also showed that lncRNA regulated PI3K-Akt signaling pathway, focal adhesion, which are very crucial and interesting findings.
- Please include in discussion the potential functional effect of akt/FAK pathway activation. Do you expect potential role metastasis/migration ?
- Since these findings are derived through pathways analysis, the authors may have included some in vitro functional studies for one of the lnRNAs eg. SLC26A4-AS1 to prove that these results are impactful.
- Results and conclusions are well stated. Appropriate controls are used.
- The authors do discuss the significance of their findings, including lncRNA NR2F1-AS1 regulating hepatocellular carcinoma.
- Please add a paragraph how these findings would affect globally in terms of translational/clinical research. Would it help to identify lnRNAs as biomarkers for patient selection? can these be targeted? Have they been studied as biomarkers of response?

---

## Round 0.2 · accepted · Accept

Please can you liaise with the office staff directly concerning the figures 2 and 3; it is important that these can be produced clearly in the on-line journal. They can discuss this with you. Thank you, meanwhile, for clarifying the paper and dealing effectively with the comments raised. I am happy to recommend this is now accepted.